# A message-passing algorithm
# for multi-agent trajectory planning

**José Bento** *
jbento@disneyresearch.com

**Nate Derbinsky**
nate.derbinsky@disneyresearch.com

**Javier Alonso-Mora**
jalonso@disneyresearch.com

**Jonathan Yedidia**
yedidia@disneyresearch.com

## Abstract

We describe a novel approach for computing collision-free *global* trajectories for $p$ agents with specified initial and final configurations, based on an improved version of the alternating direction method of multipliers (ADMM). Compared with existing methods, our approach is naturally parallelizable and allows for incorporating different cost functionals with only minor adjustments. We apply our method to classical challenging instances and observe that its computational requirements scale well with $p$ for several cost functionals. We also show that a specialization of our algorithm can be used for *local* motion planning by solving the problem of joint optimization in velocity space.

## 1 Introduction

Robot navigation relies on at least three sub-tasks: localization, mapping, and motion planning. The latter can be described as an optimization problem: compute the lowest-cost path, or *trajectory*, between an initial and final configuration. This paper focuses on trajectory planning for *multiple* agents, an important problem in robotics [1, 2], computer animation, and crowd simulation [3].

Centralized planning for multiple agents is PSPACE hard [4, 5]. To contend with this complexity, traditional multi-agent planning prioritizes agents and computes their trajectories sequentially [6], leading to suboptimal solutions. By contrast, our method plans for all agents simultaneously. Trajectory planning is also simplified if agents are non-distinct and can be dynamically assigned to a set of goal positions [1]. We consider the harder problem where robots have a unique identity and their goal positions are statically pre-specified. Both mixed-integer quadratic programming (MIQP) [7] and [more efficient, although local] sequential convex programming [8] approaches have been applied to the problem of computing collision-free trajectories for multiple agents with pre-specified goal positions; however, due to the non-convexity of the problem, these approaches, especially the former, do not scale well with the number of agents. Alternatively, trajectories may be found by sampling in their joint configuration space [9]. This approach is probabilistic and, alone, only gives asymptotic guarantees. See Appendix A for further comments on discrete search methods.

Due to the complexity of planning collision-free trajectories, *real-time* robot navigation is commonly decoupled into a global planner and a fast local planner that performs collision-avoidance. Many single-agent reactive collision-avoidance algorithms are based either on potential fields [10], which typically ignore the velocity of other agents, or "velocity obstacles" [11], which provide improved performance in dynamic environments by formulating the optimization in velocity space instead of Cartesian space. Building on an extension of the velocity-obstacles approach, recent work on centralized collision avoidance [12] computes collision-free local motions for all agents whilst maximizing a joint utility using either a computationally expensive MIQP or an efficient, though local, QP. While not the main focus of this paper, we show that a specialization of our approach

to global-trajectory optimization also applies for local-trajectory optimization, and our numerical results demonstrate improvements in both efficiency and scaling performance.

In this paper we formalize the global trajectory planning task as follows. Given $p$ agents of different radii $\{r_i\}_{i=1}^p$ with given desired initial and final positions, $\{x_i(0)\}_{i=1}^p$ and $\{x_i(T)\}_{i=1}^p$, along with a cost functional over trajectories, compute collision-free trajectories for all agents that minimize the cost functional. That is, find a set of intermediate points $\{x_i(t)\}_{i=1}^p$, $t \in (0, T)$, that satisfies the "hard" collision-free constraints that $\|x_i(t) - x_j(t)\| > r_i + r_j$, for all $i$, $j$ and $t$, and that insofar as possible, minimizes the cost functional.

The method we propose searches for a solution within the space of piece-wise linear trajectories, wherein the trajectory of an agent is completely specified by a set of positions at a fixed set of time instants $\{t_s\}_{s=0}^\eta$. We call these time instants *break-points* and they are the same for all agents, which greatly simplifies the mathematics of our method. All other intermediate points of the trajectories are computed by assuming that each agent moves with constant velocity in between break-points: if $t_1$ and $t_2 > t_1$ are consecutive break-points, then $x_i(t) = \frac{1}{t_2 - t_1}((t_2 - t)x_i(t_1) + (t - t_1)x_i(t_2))$ for $t \in [t_1, t_2]$. Along with the set of initial and final configurations, the number of interior break-points $(\eta - 1)$ is an input to our method, with a corresponding tradeoff: increasing $\eta$ yields trajectories that are more flexible and smooth, with possibly higher quality; but increasing $\eta$ enlarges the problem, leading to potentially increased computation.

The main contributions of this paper are as follows:

i) We formulate the global trajectory planning task as a decomposable optimization problem. We show how to solve the resulting sub-problems exactly and efficiently, despite their non-convexity, and how to coordinate their solutions using message-passing. Our method, based on the "three-weight" version of ADMM [13], is easily parallelized, does not require parameter tuning, and we present empirical evidence of good scalability with $p$.

ii) Within our decomposable framework, we describe different sub-problems, called *minimizers*, each ensuring the trajectories satisfy a separate criterion. Our method is flexible and can consider different combinations of minimizers. A particularly crucial minimizer ensures there are no inter-agent collisions, but we also derive other minimizers that allow for finding trajectories with minimal total energy, avoiding static obstacles, or imposing dynamic constraints, such as maximum/minimum agent velocity.

iii) We show that our method can specialize to perform local planning by solving the problem of joint optimization in velocity space [12].

Our work is among the few examples where the success of applying ADMM to find approximate solutions to a large non-convex problems can be judged with the naked eye, by the gracefulness of the trajectories found. This paper also reinforces the claim in [13] that small, yet important, modifications to ADMM can bring an order of magnitude increase in speed. We emphasize the importance of these modifications in our numerical experiments, where we compare the performance of our method using the three-weight algorithm (TWA) versus that of standard ADMM.

The rest of the paper is organized as follows. Section 2 provides background on ADMM and the TWA. Section 3 formulates the global-trajectory-planning task as an optimization problem and describes the separate blocks necessary to solve it (the mathematical details of solving these sub-problems are left to appendices). Section 4 evaluates the performance of our solution: its scalability with $p$, sensitivity to initial conditions, and the effect of different cost functionals. Section 5 explains how to implement a velocity-obstacle method using our method and compares its performance with prior work. Finally, Section 6 draws conclusions and suggests directions for future work.

## 2 Minimizers in the TWA

In this section we provide a short description of the TWA [13], and, in particular, the role of the *minimizer* building blocks that it needs to solve a particular optimization problem. Section B of the supplementary material includes a full description of the TWA.

As a small illustrative example of how the TWA is used to solve optimization problems, suppose we want to solve $\min_{x \in \mathbb{R}^3} f(x) = \min_{\{x_1, x_2, x_3\}} f_1(x_1, x_3) + f_2(x_1, x_2, x_3) + f_3(x_3)$, where $f_i(.) \in$

$\mathbb{R} \cup \{+\infty\}$. The functions can represent soft costs, for example $f_3(x_3) = (x_3 - 1)^2$, or hard equality or inequality constraints, such as $f_1(x_1, x_3) = \mathbb{J}(x_1 \le x_3)$, where we are using the notation $\mathbb{J}(.) = 0$ if $(.)$ is true or $+\infty$ if $(.)$ is false.

The TWA solves this optimization problem iteratively by passing messages on a bipartite graph, in the form of a Forney factor graph [14]: one *minimizer-node* per function $f_b$, one *equality-node* per variable $x_j$ and an edge $(b, j)$, connecting $b$ and $j$, if $f_b$ depends on $x_j$ (see Figure 1-left).

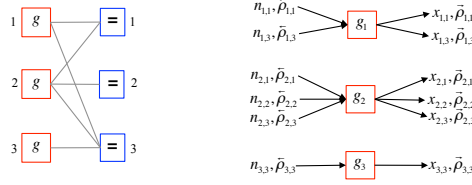

Figure 1: Left: bipartite graph, with one minimizer-node on the left for each function making up the overall objective function, and one equality-node on the right per variable in the problem. Right: The input and output variables for each minimizer block.

Apart from the first-iteration message values, and two internal parameters[1] that we specify in Section 4, the algorithm is fully specified by the behavior of the minimizers and the topology of the graph.

What does a minimizer do? The minimizer-node $g_1$, for example, solves a small optimization problem over its local variables $x_1$ and $x_3$. Without going into the full detail presented in [13] and the supplementary material, the estimates $x_{1,1}$ and $x_{1,3}$ are then combined with running sums of the differences between the minimizer estimates and the equality-node consensus estimates to obtain *messages* $m_{1,1}$ and $m_{1,3}$ on each neighboring edge that are sent to the neighboring equality-nodes along with corresponding *certainty weights*, $\overrightarrow{\rho}_{1,2}$ and $\overrightarrow{\rho}_{1,3}$. All other minimizers act similarly.

The equality-nodes receive these local messages and weights and produce consensus estimates for all variables by computing an average of the incoming messages, weighted by the incoming certainty weights $\overrightarrow{\rho}$. From these consensus estimates, *correcting messages* are computed and communicated back to the minimizers to help them reach consensus. A certainty weight for the correcting messages, $\overleftarrow{\rho}$, is also communicated back to the minimizers. For example, the minimizer $g_1$ receives correcting messages $n_{1,1}$ and $n_{1,3}$ with corresponding certainty weights $\overleftarrow{\rho}_{1,1}$ and $\overleftarrow{\rho}_{1,3}$ (see Figure 1-right).

When producing new local estimates, the $b$th minimizer node computes its local estimates $\{x_j\}$ by choosing a point that minimizes the sum of the local function $f_b$ and weighted squared distance from the incoming messages (ties are broken randomly):

$$\{x_{b,j}\}_j = g_b \left( \{n_{b,j}\}_j, \{\overleftarrow{\rho}_{b,j}^k\}_j \right) \equiv \arg \min_{\{x_j\}_j} \left[ f_b(\{x_j\}_j) + \frac{1}{2} \sum_j \overleftarrow{\rho}_{b,j} (x_j - n_{b,j})^2 \right], \quad (1)$$

where $\{\}_j$ and $\sum_j$ run over all equality-nodes connected to $b$. In the TWA, the certainty weights $\{\overrightarrow{\rho}_{b,j}\}$ that this minimizer outputs must be 0 (*uncertain*); $\infty$ (*certain*); or $\rho_0$, set to some fixed value. The logic for setting weights from minimizer-nodes depends on the problem; as we shall see, in trajectory planning problems, we only use 0 or $\rho_0$ weights. If we choose that all minimizers always output weights equal to $\rho_0$, the TWA reduces to standard ADMM; however, 0-weights allows equality-nodes to ignore inactive constraints, traversing the search space much faster.

Finally, notice that all minimizers can operate simultaneously, and the same is true for the consensus calculation performed by each equality-node. The algorithm is thus easy to parallelize.

## 3 Global trajectory planning

We now turn to describing our decomposition of the global trajectory planning optimization problem in detail. We begin by defining the variables to be optimized in our optimization problem. In

our formulation, we are *not* tracking the points of the trajectories by a continuous-time variable taking values in $[0, T]$. Rather, our variables are the positions $\{x_i(s)\}_{i \in [p]}$, where the trajectories are indexed by $i$ and break-points are indexed by a discrete variable $s$ taking values between 1 and $\eta - 1$. Note that $\{x_i(0)\}_{i \in [p]}$ and $\{x_i(\eta)\}_{i \in [p]}$ are the initial and final configuration, sets of fixed values, not variables to optimize.

### 3.1 Formulation as unconstrained optimization without static obstacles

In terms of these variables, the *non-collision* constraints[2] are

$$\|(\alpha x_i(s+1) + (1-\alpha)x_i(s)) - (\alpha x_j(s+1) + (1-\alpha)x_j(s))\| \geq r_i + r_j, \qquad (2)$$
$$\text{for all } i, j \in [p], s \in \{0, ..., \eta - 1\} \text{ and } \alpha \in [0, 1].$$

The parameter $\alpha$ is used to trace out the constant-velocity trajectories of agents $i$ and $j$ between break-points $s + 1$ and $s$. The parameter $\alpha$ has no units, it is a normalized time rather than an absolute time. If $t_1$ is the absolute time of the break-point with integer index $s$ and $t_2$ is the absolute time of the break-point with integer index $s + 1$ and $t$ parametrizes the trajectories in absolute time then $\alpha = (t - t_1)/(t_2 - t_1)$. Note that in the above formulation, absolute time does not appear, and any solution is simply a set of paths that, when travelled by each agent at constant velocity between break-points, leads to no collisions. When converting this solution into trajectories parameterized by absolute time, the break-points do not need to be chosen uniformly spaced in absolute time.

The constraints represented in (2) can be formally incorporated into an unconstrained optimization problem as follows. We search for a solution to the problem:

$$\min_{\{x_i(s)\}_{i,s}} f^{\text{cost}}(\{x_i(s)\}_{i,s}) + \sum_{s=0}^{n-1} \sum_{i>j} f^{\text{coll}}_{r_i, r_j}(x_i(s), x_i(s+1), x_j(s), x_j(s+1)), \qquad (3)$$

where $\{x_i(0)\}_p$ and $\{x_i(\eta)\}_p$ are constants rather than optimization variables, and where the function $f^{\text{cost}}$ is a function that represents some cost to be minimized (e.g. the integrated kinetic energy or the maximum velocity over all the agents) and the function $f^{\text{coll}}_{r, r'}$ is defined as,

$$f^{\text{coll}}_{r, r'}(\underline{x}, \overline{x}, \underline{x}', \overline{x}') = \mathbb{J}(\|\alpha(\overline{x} - \overline{x}') + (1 - \alpha)(\underline{x} - \underline{x}')\| \geq r + r' \ \forall \alpha \in [0, 1]). \qquad (4)$$

In this section, $\underline{x}$ and $\overline{x}$ represent the position of an arbitrary agent of radius $r$ at two consecutive break-points and $\underline{x}'$ and $\overline{x}'$ the position of a second arbitrary agent of radius $r'$ at the same break-points. In the expression above $\mathbb{J}(.)$ takes the value 0 whenever its argument, a clause, is true and takes the value $+\infty$ otherwise. Intuitively, we pay an infinite cost in $f^{\text{coll}}_{r, r'}$ whenever there is a collision, and we pay zero otherwise.

In (3) we can set $f^{\text{cost}}(.)$, to enforce a preference for trajectories satisfying specific properties. For example, we might prefer trajectories for which the total kinetic energy spent by the set of agents is small. In this case, defining $f^{\text{cost}}_C(\underline{x}, \overline{x}) = C\|\underline{x} - \overline{x}\|^2$, we have,

$$f^{\text{cost}}(\{x_i(s)\}_{i,s}) = \frac{1}{pn} \sum_{i=1}^{p} \sum_{s=0}^{n-1} f^{\text{cost}}_{C_{i,s}}(x_i(s), x_i(s+1)). \qquad (5)$$

where the coefficients $\{C_{i,s}\}$ can account for agents with different masses, different absolute-time intervals between-break points or different preferences regarding which agents we want to be less active and which agents are allowed to move faster.

More simply, we might want to exclude trajectories in which agents move faster than a certain amount, but without distinguishing among all remaining trajectories. For this case we can write,

$$f^{\text{cost}}_C(\underline{x}, \overline{x}) = \mathbb{J}(\|\overline{x} - \underline{x}\| \leq C). \qquad (6)$$

In this case, associating each break-point to a time instant, the coefficients $\{C_{i,s}\}$ in expression (5) would represent different limits on the velocity of different agents between different sections of the trajectory. If we want to force all agents to have a minimum velocity we can simply reverse the inequality in (6).

## 3.2 Formulation as unconstrained optimization with static obstacles

In many scenarios agents should also avoid collisions with static obstacles. Given two points in space, $x_L$ and $x_R$, we can forbid all agents from crossing the line segment from $x_L$ to $x_R$ by adding the following term to the function (3): $\sum_{i=1}^{p} \sum_{s=0}^{n-1} f_{x_L,x_R,r_i}^{\text{wall}}(x_i(s), x_i(s+1))$. We recall that $r_i$ is the radius of agent $i$ and

$$f_{x_L,x_R,r}^{\text{wall}}(\underline{x}, \overline{x}) = \mathbb{J}(\|(\alpha\overline{x} + (1-\alpha)\underline{x}) - (\beta x_R + (1-\beta)x_L)\| \geq r \text{ for all } \alpha, \beta \in [0,1]). \quad (7)$$

Notice that $f^{\text{coll}}$ can be expressed using $f^{\text{wall}}$. In particular,

$$f_{r,r'}^{\text{coll}}(\underline{x}, \overline{x}, \underline{x}', \overline{x}') = f_{0,0,r+r'}^{\text{wall}}(\underline{x}' - \underline{x}, \overline{x}' - \overline{x}). \quad (8)$$

We use this fact later to express the minimizer associated with agent-agent collisions using the minimizer associated with agent-obstacle collisions.

When agents move in the plane, i.e. $x_i(s) \in \mathbb{R}^2$ for all $i \in [p]$ and $s+1 \in [\eta+1]$, being able to avoid collisions with a general static line segment allows to automatically avoid collisions with multiple static obstacles of arbitrary polygonal shape. Our numerical experiments only consider agents in the plane and so, in this paper, we only describe the minimizer block for wall collision for a 2D world. In higher dimensions, different obstacle primitives need to be considered.

## 3.3 Message-passing formulation

To solve (3) using the TWA, we need to specify the topology of the bipartite graph associated with the unconstrained formulation (3) and the operation performed by every minimizer, i.e. the $\overrightarrow{\rho}$-weight update logic and $x$-variable update equations. We postpone describing the choice of initial values and internal parameters until Section 4.

We first describe the bipartite graph. To be concrete, let us assume that the cost functional has the form of (5). The unconstrained formulation (3) then tells us that the global objective function is the sum of $\eta p(p+1)/2$ terms: $\eta p(p-1)/2$ functions $f^{\text{coll}}$ and $\eta p$ functions $f_C^{\text{cost}}$. These functions involve a total of $(\eta+1)p$ variables out of which only $(\eta-1)p$ are free (since the initial and final configurations are fixed). Correspondingly, the bipartite graph along which messages are passed has $\eta p(p+1)/2$ minimizer-nodes that connect to the $(\eta+1)p$ equality-nodes. In particular, the equality-node associated with the break-point variable $x_i(s)$, $\eta > s > 0$, is connected to $2(p-1)$ different $g^{\text{coll}}$ minimizer-nodes and two different $g_C^{\text{cost}}$ minimizer-nodes. If $s = 0$ or $s = \eta$ the equality-node only connects to half as many $g^{\text{coll}}$ nodes and $g_C^{\text{cost}}$ nodes.

We now describe the different minimizers. Every minimizer basically is a special case of (1).

### 3.3.1 Agent-agent collision minimizer

We start with the minimizer associated with the functions $f^{\text{coll}}$, that we denoted by $g^{\text{coll}}$. This minimizer receives as parameters the radius, $r$ and $r'$, of the two agents whose collision it is avoiding. The minimizer takes as input a set of incoming $n$-messages, $\{\underline{n}, \overline{n}, \underline{n}', \overline{n}'\}$, and associated $\overleftarrow{\rho}$-weights, $\{\overleftarrow{\underline{\rho}}, \overleftarrow{\overline{\rho}}, \overleftarrow{\underline{\rho}}', \overleftarrow{\overline{\rho}}'\}$, and outputs a set of updated $x$-variables according to expression (9). Messages $\underline{n}$ and $\overline{n}$ come from the two equality-nodes associated with the positions of one of the agents at two consecutive break-points and $\underline{n}'$ and $\overline{n}'$ from the corresponding equality-nodes for the other agent.

$$g^{\text{coll}}(\underline{n}, \overline{n}, \underline{n}', \overline{n}', \overleftarrow{\underline{\rho}}, \overleftarrow{\overline{\rho}}, \overleftarrow{\underline{\rho}}', \overleftarrow{\overline{\rho}}', r, r') = \arg\min_{\{\underline{x}, \overline{x}, \underline{x}', \overline{x}'\}} f_{r,r'}^{\text{coll}}(\underline{x}, \overline{x}, \underline{x}', \overline{x}')$$

$$+ \frac{\overleftarrow{\underline{\rho}}}{2}\|\underline{x} - \underline{n}\|^2 + \frac{\overleftarrow{\overline{\rho}}}{2}\|\overline{x} - \overline{n}\|^2 + \frac{\overleftarrow{\underline{\rho}}'}{2}\|\underline{x}' - \underline{n}'\|^2 + \frac{\overleftarrow{\overline{\rho}}'}{2}\|\overline{x}' - \overline{n}'\|^2. \quad (9)$$

The update logic for the weights $\overrightarrow{\rho}$ for this minimizer is simple. If the trajectory from $\underline{n}$ to $\overline{n}$ for an agent of radius $r$ does not collide with the trajectory from $\underline{n}'$ to $\overline{n}'$ for an agent of radius $r'$ then set all the outgoing weights $\overrightarrow{\rho}$ to zero. Otherwise set them all to $\rho_0$. The outgoing zero weights indicate to the receiving equality-nodes in the bipartite graph that the collision constraint for this pair of agents is inactive and that the values it receives from this minimizer-node should be ignored when computing the consensus values of the receiving equality-nodes.

The solution to (9) is found using the agent-obstacle collision minimizer that we describe next.

### 3.3.2 Agent-obstacle collision minimizer

The minimizer for $f^{\text{wall}}$ is denoted by $g^{\text{wall}}$. It is parameterized by the obstacle position $\{x_L, x_R\}$ as well as the radius of the agent that needs to avoid the obstacle. It receives two $n$-messages, $\{\underline{n}, \overline{n}\}$, and corresponding weights $\{\overleftarrow{\underline{\rho}}, \overleftarrow{\overline{\rho}}\}$, from the equality-nodes associated with two consecutive positions of an agent that needs to avoid the obstacle. Its output, the $x$-variables, are defined as

$$g^{\text{wall}}(\underline{n}, \overline{n}, r, x_L, x_R, \overleftarrow{\underline{\rho}}, \overleftarrow{\overline{\rho}}) = \arg\min_{\{\underline{x}, \overline{x}\}} f^{\text{wall}}_{x_L, x_R, r}(\underline{x}, \overline{x}) + \frac{\overleftarrow{\underline{\rho}}}{2}\|\underline{x} - \underline{n}\|^2 + \frac{\overleftarrow{\overline{\rho}}}{2}\|\overline{x} - \overline{n}\|^2. \quad (10)$$

When agents move in the plane (2D), this minimizer can be solved by reformulating the optimization in (10) as a mechanical problem involving a system of springs that we can solve exactly and efficiently. This reduction is explained in the supplementary material in Section D and the solution to the mechanical problem is explained in Section I.

The update logic for the $\overrightarrow{\rho}$-weights is similar to that of the $g^{\text{coll}}$ minimizer. If an agent of radius $r$ going from $\underline{n}$ and $\overline{n}$ does not collide with the line segment from $x_L$ to $x_R$ then set all outgoing weights to zero because the constraint is inactive; otherwise set all the outgoing weights to $\rho_0$.

Notice that, from (8), it follows that the agent-agent minimizer $g^{\text{coll}}$ can be expressed using $g^{\text{wall}}$. More concretely, as proved in the supplementary material, Section C,

$$g^{\text{coll}}(\underline{n}, \overline{n}, \underline{n}', \overline{n}', \overleftarrow{\underline{\rho}}, \overleftarrow{\overline{\rho}}, \overleftarrow{\underline{\rho}}', \overleftarrow{\overline{\rho}}', r, r') = M_2 g^{\text{wall}}\left(M_1.\{\underline{n}, \overline{n}, \underline{n}', \overline{n}', \overleftarrow{\underline{\rho}}, \overleftarrow{\overline{\rho}}, \overleftarrow{\underline{\rho}}', \overleftarrow{\overline{\rho}}', r, r'\}\right),$$

for a constant rectangular matrix $M_1$ and a matrix $M_2$ that depend on $\{\underline{n}, \overline{n}, \underline{n}', \overline{n}', \overleftarrow{\underline{\rho}}, \overleftarrow{\overline{\rho}}, \overleftarrow{\underline{\rho}}', \overleftarrow{\overline{\rho}}'\}$.

### 3.3.3 Minimum energy and maximum (minimum) velocity minimizer

When $f^{\text{cost}}$ can be decomposed as in (5), the minimizer associated with the functions $f^{\text{cost}}$ is denoted by $g^{\text{cost}}$ and receives as input two $n$-messages, $\{\underline{n}, \overline{n}\}$, and corresponding weights, $\{\overleftarrow{\underline{\rho}}, \overleftarrow{\overline{\rho}}\}$. The messages come from two equality-nodes associated with two consecutive positions of an agent. The minimizer is also parameterized by a cost factor $c$. It outputs a set of updated $x$-messages defined as

$$g^{\text{cost}}(\underline{n}, \overline{n}, \overleftarrow{\underline{\rho}}, \overleftarrow{\overline{\rho}}, c) = \arg\min_{\{\underline{x}, \overline{x}\}} f^{\text{cost}}_c(\underline{x}, \overline{x}) + \frac{\overleftarrow{\underline{\rho}}}{2}\|\underline{x} - \underline{n}\|^2 + \frac{\overleftarrow{\overline{\rho}}}{2}\|\overline{x} - \overline{n}\|^2. \quad (11)$$

The update logic for the $\overrightarrow{\rho}$-weights of the minimum energy minimizer is very simply: always set all outgoing weights $\overrightarrow{\rho}$ to $\rho_0$. The update logic for the $\overrightarrow{\rho}$-weights of the maximum velocity minimizer is the following. If $\|\underline{n} - \overline{n}\| \leq c$ set all outgoing weights to zero. Otherwise, set them to $\rho_0$. The update logic for the minimum velocity minimizer is similar. If $\|\underline{n} - \overline{n}\| \geq c$, set all the $\overrightarrow{\rho}$-weights to zero. Otherwise set them to $\rho_0$.

The solution to the minimum energy, maximum velocity and minimum velocity minimizer is written in the supplementary material in Sections E, F, and G respectively.

## 4 Numerical results

We now report on the performance of our algorithm (see Appendix J for an important comment on the anytime properties of our algorithm). Note that the lack of open-source scalable algorithms for global trajectory planning in the literature makes it difficult to benchmark our performance against other methods. Also, in a paper it is difficult to appreciate the gracefulness of the discovered trajectory optimizations, so we include a video in the supplementary material that shows final optimized trajectories as well as intermediate results as the algorithm progresses for a variety of additional scenarios, including those with obstacles. All the tests described here are for agents in a two-dimensional plane. All tests but the last were performed using six cores of a 3.4GHz i7 CPU.

The different tests did not require any special tuning of parameters. In particular, the step-size in [13] (their $\alpha$ variable) is always $0.1$. In order to quickly equilibrate the system to a reasonable set of variables and to wash out the importance of initial conditions, the default weight $\rho_0$ was set equal to a small value ($\eta p \times 10^{-5}$) for the first 20 iterations and then set to 1 for all further iterations.

The first test considers scenario CONF1: $p$ (even) agents of radius $r$, equally spaced around on a circle of radius $R$, are each required to exchange position with the corresponding antipodal agent, $r = (5/4)R\sin(\pi/2(p-4))$. This is a classical difficult test scenario because the straight line motion of all agents to their goal would result in them all colliding in the center of the circle. We compare the convergence time of the TWA with a similar version using standard ADMM to perform the optimizations. In this test, the algorithm's initial value for each variable in the problem was set to the corresponding initial position of each agent. The objective is to minimize the total kinetic energy ($C$ in the energy minimizer is set to 1). Figure 2-left shows that the TWA scales better with $p$ than classic ADMM and typically gives an order of magnitude speed-up. Please see Appendix K for a further comment on the scaling of the convergence time of ADMM and TWA with $p$.

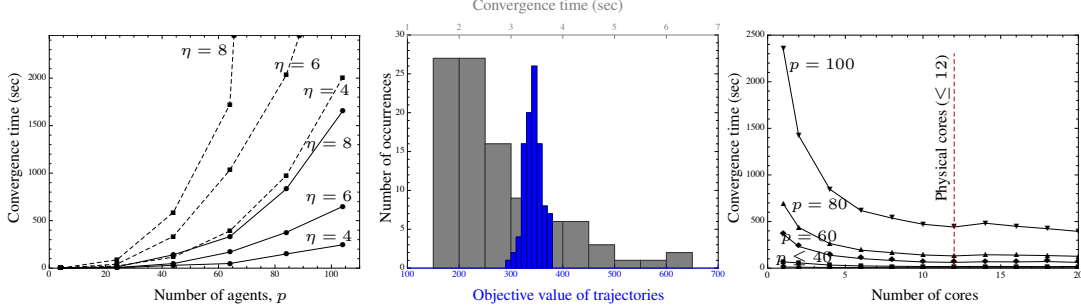

Figure 2: Left: Convergence time using standard ADMM (dashed lines) and using TWA (solid lines). Middle: Distribution of total energy and time for convergence with random initial conditions ($p = 20$ and $\eta = 5$). Right: Convergence time using a different number of cores ($\eta = 5$).

The second test for CONF1 analyzes the sensitivity of the convergence time and objective value when the variables' value at the first iteration are chosen uniformly at random in the smallest space-time box that includes the initial and final configuration of the robots. Figure 2-middle shows that, although there is some spread on the convergence time, our algorithm seems to reliably converge to relatively similar-cost local minima (other experiments show that the objective value of these minima is around 5 times smaller than that found when the algorithm is run using only the collision avoidance minimizers without a kinetic energy cost term). As would be expected, the precise trajectories found vary widely between different random runs.

Still for CONF1, and fixed initial conditions, we parallelize our method using several cores of a 2.66GHz i7 processor and a very primitive scheduling/synchronization scheme. Although this scheme does not fully exploit parallelization, Figure 2-right does show a speed-up as the number of cores increases and the larger $p$ is, the greater the speed-up. We stall when we reach the twelve physical cores available and start using virtual cores.

Finally, Figure 3-left compares the convergence time to optimize the total energy with the time to simply find a feasible (i.e. collision-free) solution. The agents initial and final configuration is randomly chosen in the plane (CONF2). Error bars indicate $\pm$ one standard deviation. Minimizing the kinetic energy is orders of magnitude computationally more expensive than finding a feasible solution, as is clear from the different magnitude of the left and right scale of Figure 3-left.

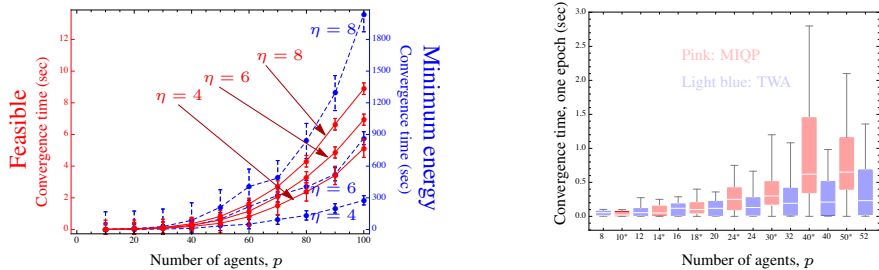

Figure 3: Left: Convergence time when minimizing energy (blue scale/dashed lines) and to simply find a feasible solution (red scale/solid lines). Right: (For Section 5). Convergence-time distribution for each epoch using our method (blue bars) and using the MIQP of [12] (red bars and star-values).

## 5 Local trajectory planning based on velocity obstacles

In this section we show how the joint optimization presented in [12], which is based on the concept of velocity obstacles [11] (VO), can be also solved via the message-passing TWA. In VO, given the current position $\{x_i(0)\}_{i\in[p]}$ and radius $\{r_i\}$ of all agents, a new velocity command is computed jointly for all agents minimizing the distance to their preferred velocity $\{v_i^{\text{ref}}\}_{i\in[p]}$. This new velocity command must guarantee that the trajectories of all agents remain collision-free for at least a time horizon $\tau$. New collision-free velocities are computed every $\alpha\tau$ seconds, $\alpha < 1$, until all agents reach their final configuration. Following [12], and assuming an obstacle-free environment and first order dynamics, the collision-free velocities are given by,

$$\underset{\{v_i\}_{i\in[p]}}{\text{minimize}} \sum_{i\in[p]} C_i\|v_i - v_i^{\text{ref}}\|^2 \text{ s.t. } \|(x_i(0) + v_i t) - (x_j(0) + v_j t)\| \geq r_i + r_j \ \ \forall \ i \in [p], t \in [0, \tau].$$

Since the velocities $\{v_i\}_{i\in[p]}$ are related linearly to the final position of each object after $\tau$ seconds, $\{x_i(\tau)\}_{i\in[p]}$, a simple change of variables allows us to reformulate the above problem as,

$$\underset{\{x_i\}_{i\in[p]}}{\text{minimize}} \sum_{i\in[p]} C_i'\|x_i - x_i^{\text{ref}}\|^2$$

$$\text{s.t. } \|(1 - \alpha)(x_i(0) - x_j(0)) + \alpha(x_i - x_j)\| \geq r_i + r_j \ \forall \ j > i \in [p], \alpha \in [0, 1] \tag{12}$$

where $C_i' = C_i/\tau^2$, $x_i^{\text{ref}} = x_i(0) + v_i^{\text{ref}}\tau$ and we have dropped the $\tau$ in $x_i(\tau)$. The above problem, extended to account for collisions with the static line segments $\{x_{Rk}, x_{Lk}\}_k$, can be formulated in an unconstrained form using the functions $f^{\text{cost}}$, $f^{\text{coll}}$ and $f^{\text{wall}}$. Namely,

$$\underset{\{x_i\}_i}{\min} \sum_{i\in[p]} f_{C_i'}^{\text{cost}}(x_i, x_i^{\text{ref}}) + \sum_{i>j} f_{r_i, r_j}^{\text{coll}}(x_i(0), x_i, x_j(0), x_j) + \sum_{i\in[p]} \sum_k f_{x_{Rk}, x_{Lk}, r_i}^{\text{wall}}(x_i(0), x_i). \tag{13}$$

Note that $\{x_i(0)\}_i$ and $\{x_i^{\text{ref}}\}_i$ are constants, not variables being optimized. Given this formulation, the TWA can be used to solve the optimization. All corresponding minimizers are special cases of minimizers derived in the previous section for global trajectory planning (see Section H in the supplementary material for details). Figure 3-right shows the distribution of the time to solve (12) for CONF1. We compare the mixed integer quadratic programming (MIQP) approach from [12] with ours. Our method finds a local minima of exactly (13), while [12] finds a global minima of an approximation to (13). Specifically, [12] requires approximating the search domain by hyperplanes and an additional branch-and-bound algorithm while ours does not. Both approaches use a mechanism for breaking the symmetry from CONF1 and avoid deadlocks: theirs uses a preferential rotation direction for agents, while we use agents with slightly different $C$ coefficients in their energy minimizers ($C_{i^{th} \text{ agent}} = 1 + 0.001i$). Both simulations were done on a single 2.66GHz core. The results show the order of magnitude is similar, but, because our implementation is done in Java while [12] uses Matlab-mex interface of CPLEX 11, the results are not exactly comparable.

## 6 Conclusion and future work

We have presented a novel algorithm for global and local planning of the trajectory of multiple distinct agents, a problem known to be hard. The solution is based on solving a non-convex optimization problem using TWA, a modified ADMM. Its similarity to ADMM brings scalability and easy parallelization. However, using TWA improves performance considerably. Our implementation of the algorithm in Java on a regular desktop computer, using a basic scheduler/synchronization over its few cores, already scales to hundreds of agents and achieves real-time performance for local planning.

The algorithm can flexibly account for obstacles and different cost functionals. For agents in the plane, we derived explicit expressions that account for static obstacles, moving obstacles, and dynamic constraints on the velocity and energy. Future work should consider other restrictions on the smoothness of the trajectory (e.g. acceleration constraints) and provide fast solvers to our minimizers for agents in 3D.

The message-passing nature of our algorithm hints that it might be possible to adapt our algorithm to do planning in a decentralized fashion. For example, minimizers like $g^{\text{coll}}$ could be solved by message exchange between pairs of agents within a maximum communication radius. It is an open problem to build a practical communication-synchronization scheme for such an approach.

## Footnotes

*This author would like to thank Emily Hupf and Noa Ghersin for their support while writing this paper.

[1]These are the step-size and $\rho_0$ constants. See Section B in the supplementary material for more detail.

[2]We replaced the strict inequality in the condition for non-collision by a simple inequality "$\geq$" to avoid technicalities in formulating the optimization problem. Since the agents are round, this allows for a single point of contact between two agents and does not reduce practical relevance.

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
