[Supplementary Material]

# Supplementary material for "A message-passing algorithm for multi-agent trajectory planning"

This document gives details on the message-passing algorithm we use and how to implement all minimizers described in the paper. It also includes some more comments on our numerical results and related literature. A short movie clip showing the behaviour of our algorithm can be found at *http://youtu.be/yuGCkVT8Bew*

## A    Comment on related literature

$A^*$-search based methods and sampling-based methods require exploring a continuous domain using discrete graph structures. For problems with many degrees of freedom or complex kinematic and dynamic constraints, as when dealing with multiple agents or manipulators, fixed-grid search methods are impractical. Alternatively, exploration can be done using sampling algorithms with proved asymptotic convergence to the optimal solution [15]. However, as the dimensionality of the configuration space increases, the convergence rate degrades and the local planners required by the exploration loop become harder to implement. In addition, as pointed out in [9], sampling algorithms cannot easily produce solutions where multiple agents move in tight spaces, like in CONF1 with obstacles. Some of the disadvantages of using discrete random search structures are even visible in extremely simple scenarios. For example, for a single holonomic agent that needs to move as quickly as possible between two points in free-space, [15] require around 10000 samples on their RRT* method to find something close to the shortest-path solution. For our algorithm this is a trivial scenario: it outputs the optimal straight-line solution in 200 iterations and 37 msecs. in our Java implementation.

## B    Full description of the improved three-weight message-passing algorithm of [13]

First we give a self-contained (complete) description of the three-weight algorithm TWA from [13]. Their method is an improvement of the alternating direction method of multipliers (ADMM) [3]. Assume we want to solve

$$\min_{x \in \mathbb{R}^d} \sum_{b=1}^{l} f_b(x_{\partial b}), \tag{14}$$

where the set $x_{\partial b} = \{x_j : j \in \partial b\}$ is a vector obtained by considering the subset of entries of $x$ with index in $\partial b \subseteq [d]$. The functions $f_b$ do not need to be convex or smooth for the algorithm to be well-defined. However, the algorithm is only guaranteed to find the global minimum under convexity [22].

Start by forming the following bipartite graph consisting of minimizer-nodes and equality-nodes. Create one minimizer-node, labeled "$g$", per function $f_b$ and one equality-node, labeled "=", per variable $x_j$. There are $l$ minimizer-nodes and $d$ equality-nodes in total. If function $f_b$ depends on variable $x_j$, create an edge $(b, j)$ connecting $b$ and $j$ (see Figure B for a general representation).

The algorithm in [13] works by repetitively updating seven kind of variables. These can be listed as follows. Every equality-node $j$ has a corresponding variable $z_j$. Every edge $(b, j)$ from minimizer-node $b$ to equality-node $j$ has a corresponding variable $x_{b,j}$ , variable $u_{b,j}$, message $n_{b,j}$, message $m_{b,j}$, weight $\overrightarrow{\rho}_{b,j}$ and weight $\overleftarrow{\rho}_{b,j}$.

To start the method, one specifies the initial values $\{z_j^0\}$, $\{u_{b,j}^0\}$ and $\{\overleftarrow{\rho}_{b,j}^0\}$. Then, at every iteration $k$, repeat the following.

Figure 4: Bipartite graph used by the message-passing algorithm to solve (14). The algorithm works by updating seven kind of variables, also shown in picture.

Figure 5: Left: Special update-rules for variables $u$. Right: Update rule for variables $\overleftarrow{\rho}$.

1. Construct the $n$-message for every edge $(b, j)$ as $n_{b,j}^k = z_j^k - u_{b,j}^k$.

2. Update the $x$-variables for every edge $(b, j)$. All $x$-variables associated to edges incident on the same minimizer-node $b$, i.e. $\{x_{b,j}^k\}_{j\in\partial b}$, are updated simultaneously by the minimizer $g_b$ associated to the function $f_b$,

$$\{x_{b,j}^{k+1}\}_j = g_b\left(\{n_{b,j}^k\}_j, \{\overleftarrow{\rho}_{b,j}^k\}_j\right)$$

$$\equiv \arg\min_{\{\tilde{x}_{b,j}^k\}_j} f_b(\{\tilde{x}_{b,j}^k\}_j) + \frac{1}{2}\sum_j \overleftarrow{\rho}_{b,j}^k (\tilde{x}_{b,j}^k - n_{b,j}^k)^2.$$

   In the expression above we write $\{\}_j$ and $\sum_j$ for $\{\}_{j\in\partial b}$ and $\sum_{j\in\partial b}$ respectively. If the $\arg\min$ returns a set with more than one element, choose a value uniformly at random from this set.

3. Compute the outgoing weights for all edges $(b, j)$. All weights associated to edges leaving the same minimizer-node $b$, i.e. $\{\overrightarrow{\rho}_{b,j}^k\}_{j\in\partial b}$, are updated simultaneously according to a user-defined logic that should depend on the problem defined by equation (14). This logic can assign three possible values for the weights: $\overrightarrow{\rho}_{b,j}^k = 0$, $\overrightarrow{\rho}_{b,j}^k = \rho_0$ or $\overrightarrow{\rho}_{b,j}^k = \infty$, where $\rho_0 > 0$ is some pre-specified constant. The idea is to use these three values to inform the equality-nodes of how certain the minimizer-node $b$ is that the current variable $x_{b,j}^k$ is the optimal value for variable $x_j$ in equation (14). A weight of $\infty$ is used to signal total certainty, 0 for no certainty at all and $\rho_0$ for all the other scenarios. Later, equality-nodes average the information coming from the minimizer-nodes by these weights to update the consensus variables $z$.

4. Construct the $m$-messages for every edge $(b, j)$ as $m_{b,j}^k = x_{b,j}^{k+1} + u_{b,j}^k$.

5. Update the $z$-variable for every equality-node $j$ as

$$z_j^{k+1} = \frac{\sum_{b\in\partial j} \overrightarrow{\rho}_{b,j}^k m_{b,j}^k}{\sum_{b\in\partial j} \overrightarrow{\rho}_{b,j}^k}.$$

The set $\partial j$ contains all the minimizer-nodes that connect to equality-node $j$. If all weights $\{\overrightarrow{\rho}^{\,k}_{b,j}\}_{b\in\partial j}$ are zero in the previous expression, treat them as 1.

6. Compute the updated weights $\overleftarrow{\rho}^{\,k+1}$. Weights leaving the same equality-node $j$, i.e. $\{\overleftarrow{\rho}^{\,k+1}_{b,j}\}_{b\in\partial j}$, are computed simultaneously. The update logic is described in Figure 5-right. According to which of the three distinct scenarios $\overrightarrow{\rho}^{\,k}$ falls, $\overleftarrow{\rho}^{\,k+1}$ is uniquely determined. This logic again assigns three possible values for the weights $\{0,\rho_0,\infty\}$.

7. Update the $u$-variables for all edges. If edge $(b,j)$ has $\overrightarrow{\rho}^{\,k}_{b,j}=\rho_0$ and $\overleftarrow{\rho}^{\,k+1}_{b,j}=\rho_0$ then $u^{k+1}_{b,j}=u^{k}_{b,j}+(\alpha/\rho_0)(x^{k+1}_{b,j}-z^{k+1}_j)$, where $\alpha$ is a pre-specified constant. Otherwise, choose $u^{k+1}_{b,j}$ according to the three scenarios described in Figure 5-left, depending on the weights $\overrightarrow{\rho}$ and $\overleftarrow{\rho}$.

In short, given $\rho_0$, $\alpha$, the initial values $\{z^0_j\}$, $\{u^0_{b,j}\}$ and $\{\overleftarrow{\rho}^{\,0}_{b,j}\}$, all the minimizers $\{g_b\}$, with corresponding update logic for $\overrightarrow{\rho}$, and the bipartite graph, the method is completely specified. If all weights, $\overrightarrow{\rho}$ and $\overleftarrow{\rho}$, are set to $\rho_0$ across all iterations, the described method reduces to classical ADMM , interpreted as a message-passing algorithm. Finally, notice that at each time step $k$, all the variables associated with each edge can be updated in parallel. In particular, the update of the $x$-variables, usually the most expensive operation, can be parallelized.

## C   Agent-agent collision minimizer

Here we give the details of how to write the agent-agent collision minimizer, $g^{\text{coll}}$ using the agent-obstacle minimizer $g^{\text{wall}}$ of equation (10).

First recall that $f^{\text{coll}}_{r,r'}(\underline{x},\overline{x},\underline{x}',\overline{x}')=f^{\text{wall}}_{0,0,r+r'}(\underline{x}'-\underline{x},\overline{x}'-\overline{x})$. Then, rewrite (9) as,

$$g^{\text{coll}}(\underline{n},\overline{n},\underline{n}',\overline{n}',\overleftarrow{\underline{\rho}},\overleftarrow{\overline{\rho}},\overleftarrow{\underline{\rho}}',\overleftarrow{\overline{\rho}}',r,r')=\arg\min_{\{\underline{x},\overline{x},\underline{x}',\overline{x}'\}}\Bigg[f^{\text{wall}}_{0,0,r+r'}(\underline{x}'-\underline{x},\overline{x}'-\overline{x})$$
$$+\frac{\overleftarrow{\underline{\rho}}}{2}\|\underline{x}-\underline{n}\|^2+\frac{\overleftarrow{\overline{\rho}}}{2}\|\overline{x}-\overline{n}\|^2+\frac{\overleftarrow{\underline{\rho}}'}{2}\|\underline{x}'-\underline{n}'\|^2+\frac{\overleftarrow{\overline{\rho}}'}{2}\|\overline{x}'-\overline{n}'\|^2\Bigg]. \tag{15}$$

Now introduce the following variables $\underline{v}=\underline{x}'-\underline{x}$, $\underline{u}=\underline{x}'+\underline{x}$, $\overline{v}=\overline{x}'-\overline{x}$ and $\overline{u}=\overline{x}'+\overline{x}$. The function being minimized in (15) can be written as

$$f^{\text{wall}}_{0,0,r+r'}(\underline{v},\overline{v})+\frac{\overleftarrow{\underline{\rho}}}{2}\left\|\frac{\underline{u}-\underline{v}}{2}-\underline{n}\right\|^2+\frac{\overleftarrow{\overline{\rho}}}{2}\left\|\frac{\overline{u}-\overline{v}}{2}-\overline{n}\right\|^2$$
$$+\frac{\overleftarrow{\underline{\rho}}'}{2}\left\|\frac{\underline{u}+\underline{v}}{2}-\underline{n}'\right\|^2+\frac{\overleftarrow{\overline{\rho}}'}{2}\left\|\frac{\overline{u}+\overline{v}}{2}-\overline{n}'\right\|^2. \tag{16}$$

Now notice that we can write,

$$\frac{\overleftarrow{\underline{\rho}}}{2}\left\|\frac{\underline{u}-\underline{v}}{2}-\underline{n}\right\|^2+\frac{\overleftarrow{\underline{\rho}}'}{2}\left\|\frac{\underline{u}+\underline{v}}{2}-\underline{n}'\right\|^2=\frac{\overleftarrow{\underline{\rho}}}{8}\|\underline{u}-\underline{v}-2\underline{n}\|^2+\frac{\overleftarrow{\underline{\rho}}'}{8}\|\underline{u}+\underline{v}-2\underline{n}'\|^2$$
$$=\frac{\overleftarrow{\underline{\rho}}}{8}(\|\underline{u}\|^2+\|\underline{v}+2\underline{n}\|^2-2\langle\underline{u},\underline{v}+2\underline{n}\rangle)+\frac{\overleftarrow{\underline{\rho}}'}{8}(\|\underline{u}\|^2+\|\underline{v}-2\underline{n}'\|^2+2\langle\underline{u},\underline{v}-2\underline{n}'\rangle)$$
$$=\frac{\overleftarrow{\underline{\rho}}+\overleftarrow{\underline{\rho}}'}{8}\|\underline{u}\|^2+2\left\langle\underline{u},\frac{\overleftarrow{\underline{\rho}}'-\overleftarrow{\underline{\rho}}}{8}\underline{v}-\frac{\overleftarrow{\underline{\rho}}\,\underline{n}+\overleftarrow{\underline{\rho}}'\underline{n}'}{4}\right\rangle+\frac{\overleftarrow{\underline{\rho}}}{8}\|\underline{v}+2\underline{n}\|^2+\frac{\overleftarrow{\underline{\rho}}'}{8}\|\underline{v}-2\underline{n}'\|^2$$
$$=\frac{\overleftarrow{\underline{\rho}}+\overleftarrow{\underline{\rho}}'}{8}\left\|\underline{u}-\left(\frac{\overleftarrow{\underline{\rho}}-\overleftarrow{\underline{\rho}}'}{\overleftarrow{\underline{\rho}}+\overleftarrow{\underline{\rho}}'}\underline{v}+\frac{2(\overleftarrow{\underline{\rho}}\,\underline{n}+\overleftarrow{\underline{\rho}}'\underline{n}')}{\overleftarrow{\underline{\rho}}+\overleftarrow{\underline{\rho}}'}\right)\right\|^2+\frac{\overleftarrow{\underline{\rho}}+\overleftarrow{\underline{\rho}}'}{8}\left\|\underline{v}-\frac{2(\overleftarrow{\underline{\rho}}'\underline{n}'-\overleftarrow{\underline{\rho}}\,\underline{n})}{\overleftarrow{\underline{\rho}}+\overleftarrow{\underline{\rho}}'}\right\|^2$$
$$+C(\underline{n},\overleftarrow{\underline{\rho}},\underline{n}',\overleftarrow{\underline{\rho}}'),$$

where $C(\underline{n},\overleftarrow{\underline{\rho}},\underline{n}',\overleftarrow{\underline{\rho}}')$ is a constant that depends on the variables $\{\underline{n},\overleftarrow{\underline{\rho}},\underline{n}',\overleftarrow{\underline{\rho}}'\}$. A similar manipulation can be done to $\frac{\overleftarrow{\overline{\rho}}}{2}\left\|\frac{\overline{u}-\overline{v}}{2}-\overline{n}\right\|^2+\frac{\overleftarrow{\overline{\rho}}'}{2}\left\|\frac{\overline{u}+\overline{v}}{2}-\overline{n}'\right\|^2$. Therefore, the expression (16) can be

rewritten as

$$f^{\text{wall}}_{0,0,r+r'}(\underline{v},\overline{v}) + \frac{\overleftarrow{\rho}+\overleftarrow{\rho}'}{8}\left\|\underline{v} - \frac{2(\overleftarrow{\rho}'\underline{n}'-\overleftarrow{\rho}\,\underline{n})}{\overleftarrow{\rho}+\overleftarrow{\rho}'}\right\|^2 + \frac{\overrightarrow{\rho}+\overrightarrow{\rho}'}{8}\left\|\overline{v}-\frac{2(\overrightarrow{\rho}'\overline{n}'-\overrightarrow{\rho}\,\overline{n})}{\overrightarrow{\rho}+\overrightarrow{\rho}'}\right\|^2$$

$$+\frac{\overleftarrow{\rho}+\overleftarrow{\rho}'}{8}\left\|\underline{u}-\left(\frac{\overleftarrow{\rho}-\overleftarrow{\rho}'}{\overleftarrow{\rho}+\overleftarrow{\rho}'}\underline{v}+\frac{2(\overleftarrow{\rho}\,\underline{n}+\overleftarrow{\rho}'\underline{n}')}{\overleftarrow{\rho}+\overleftarrow{\rho}'}\right)\right\|^2$$

$$+\frac{\overrightarrow{\rho}+\overrightarrow{\rho}'}{8}\left\|\overline{u}-\left(\frac{\overrightarrow{\rho}-\overrightarrow{\rho}'}{\overrightarrow{\rho}+\overrightarrow{\rho}'}\overline{v}+\frac{2(\overrightarrow{\rho}\,\overline{n}+\overrightarrow{\rho}'\overline{n}')}{\overrightarrow{\rho}+\overrightarrow{\rho}'}\right)\right\|^2 + C(\underline{n},\underline{n}',\overline{n},\overline{n}',\overleftarrow{\rho},\overleftarrow{\rho}',\overrightarrow{\rho},\overrightarrow{\rho}'), \quad (17)$$

where $C(\underline{n},\underline{n}',\overline{n},\overline{n}',\overleftarrow{\rho},\overleftarrow{\rho}',\overrightarrow{\rho},\overrightarrow{\rho}')$ is a constant that depends on the variables $\underline{n}$, $\underline{n}'$, $\overline{n}$, $\overline{n}'$, $\overleftarrow{\rho}$, $\overleftarrow{\rho}'$, $\overrightarrow{\rho}$ and $\overrightarrow{\rho}'$.

Let $\{\underline{v}^*,\overline{v}^*,\underline{u}^*,\overline{u}^*\}$ be a set of values that minimizes equation (17). We have,

$$\{\underline{v}^*,\overline{v}^*\}\in g^{\text{wall}}\left(\frac{2(\overleftarrow{\rho}'\underline{n}'-\overleftarrow{\rho}\,\underline{n})}{\overleftarrow{\rho}+\overleftarrow{\rho}'},\frac{2(\overrightarrow{\rho}'\overline{n}'-\overrightarrow{\rho}\,\overline{n})}{\overrightarrow{\rho}+\overrightarrow{\rho}'},r+r',0,0,\frac{\overleftarrow{\rho}+\overleftarrow{\rho}'}{4},\frac{\overrightarrow{\rho}+\overrightarrow{\rho}'}{4}\right), \quad (18)$$

$$\{\underline{u}^*,\overline{u}^*\}=\left\{\frac{\overleftarrow{\rho}-\overleftarrow{\rho}'}{\overleftarrow{\rho}+\overleftarrow{\rho}'}\underline{v}^*+\frac{2(\overleftarrow{\rho}\,\underline{n}+\overleftarrow{\rho}'\underline{n}')}{\overleftarrow{\rho}+\overleftarrow{\rho}'},\frac{\overrightarrow{\rho}-\overrightarrow{\rho}'}{\overrightarrow{\rho}+\overrightarrow{\rho}'}\overline{v}^*+\frac{2(\overrightarrow{\rho}\,\overline{n}+\overrightarrow{\rho}'\overline{n}')}{\overrightarrow{\rho}+\overrightarrow{\rho}'}\right\}. \quad (19)$$

We can now produce a set of values that satisfy

$$\{\underline{x}^*,\overline{x}^*,\underline{x}'^*,\overline{x}'^*\}\in g^{\text{coll}}(\underline{n},\overline{n},\underline{n}',\overline{n}',\overleftarrow{\rho},\overrightarrow{\rho},\overleftarrow{\rho}',\overrightarrow{\rho}',r,r')$$

using the following relation,

$$\{\underline{x}^*,\overline{x}^*,\underline{x}'^*,\overline{x}'^*\}=\left\{\frac{\underline{u}^*-\underline{v}^*}{2},\frac{\overline{u}^*-\overline{v}^*}{2},\frac{\underline{v}^*+\underline{u}^*}{2},\frac{\overline{u}^*+\overline{v}^*}{2}\right\}.$$

In fact, all values $\{\underline{x}^*,\overline{x}^*,\underline{x}'^*,\overline{x}'^*\}\in g^{\text{coll}}(\underline{n},\overline{n},\underline{n}',\overline{n}',\overleftarrow{\rho},\overrightarrow{\rho},\overleftarrow{\rho}',\overrightarrow{\rho}',r,r')$ can be obtained from some $\{\underline{v}^*,\overline{v}^*,\underline{u}^*,\overline{u}^*\}$ that minimizes equation (17). In other words, the minimizer $g^{\text{coll}}$ can be expressed in terms of the minimizer $g^{\text{wall}}$ by means of a linear transformation.

Minimizers can receive zero-weight messages $\overleftarrow{\rho}$ from their neighboring equality nodes. In (18) and (19), this can lead to indeterminacies. We address this as follows. If $\overleftarrow{\rho}$ and $\overleftarrow{\rho}'$ are simultaneously zero then we compute (18) and (19) in the limit when $\overleftarrow{\rho}=\overleftarrow{\rho}'\to 0^+$. When implementing this on software, we simply replace them by small equal values. The fact that $\overleftarrow{\rho}=\overleftarrow{\rho}'$ resolves the indeterminacies in the fractions and taking the limit to zero from above guarantees that the wall minimizer $g^{\text{wall}}$, that is solved using a mechanical analogy involving springs, is well behaved (See Section I). If $\overrightarrow{\rho}$ and $\overrightarrow{\rho}'$ are simultaneously zero, we perform a similar operation.

## D   Agent-obstacle collision minimizer

In Section C we expressed the agent-agent collision minimizer by applying a linear transformation to the agent-obstacle collision minimizer. Now we show how the agent-obstacle minimizer can be posed as a classical mechanical problem involving a system of springs. Although the relationship in Section C holds in general, the transformation presented in this section holds only when the agents move in the plane, i.e. $x_i(s)\in\mathbb{R}^2\;\;\forall s,i$. Similar transformations should hold in higher dimensions.

When the obstacle is a line-segment $[x_L,x_R]$, the agent-obstacle minimizer (10) solves the following non-convex optimization problem,

$$\underset{\{\underline{x},\overline{x}\}}{\text{minimize}}\quad\left[\frac{\overleftarrow{\rho}}{2}\|\underline{x}-\underline{n}\|^2+\frac{\overrightarrow{\rho}}{2}\|\overline{x}-\overline{n}\|^2\right] \tag{20}$$

$$\text{s.t. }\|(\alpha\overline{x}+(1-\alpha)\underline{x})-(\beta x_R+(1-\beta)x_L)\|\geq r\text{ for all }\alpha,\beta\in[0,1]. \tag{21}$$

Observe that the term $\frac{\overleftarrow{\rho}}{2}\|\underline{x}-\underline{n}\|^2$ equals the energy of a spring with zero rest-length and elastic coefficient $\overleftarrow{\rho}$ whose end points are at positions $\underline{x}$ and $\underline{n}$. The same interpretation applies for the

second term in (20). With this interpretation in mind, the non-convex constraint (21) means that the line from $\underline{x}$ to $\overline{x}$ cannot cross the region swept out by a circle of radius $r$ that moves from $x_L$ to $x_R$. We call this region $\mathcal{R}$. Figure 6-left shows a feasible solution and an unfeasible solution under this interpretation. When the line from $\underline{n}$ to $\overline{n}$ does not cross $\mathcal{R}$, the solution of (20)-(21) is

Figure 6: Left: Feasible solution (blue) and unfeasible solution (red). Right: Two different feasible configurations of the springs-slab system. Each represented in different color.

$\underline{x} = \underline{n}$ and $\overline{x} = \underline{n}$. In general however, $\underline{x}$ and $\overline{x}$ adopt the minimum energy configuration of a system with two zero rest-length springs, with end points $(\underline{n}, \underline{x})$ and $(\overline{n}, \overline{x})$ and elastic coefficients $\overleftarrow{\underline{\rho}}$ and $\overleftarrow{\overline{\rho}}$, and with a hard extensible slab, connecting $\underline{x}$ to $\overline{x}$, that cannot go over region $\mathcal{R}$. The slab can be extended without spending any energy. Figure 6-right shows two feasible configurations of the system of springs and slab when $\underline{x} = \underline{n}$ and $\overline{x} = \overline{n}$ cannot be a feasible solution.

It is possible that this minimizer receives two zero-weight messages from its neighboring equality nodes, i.e., $\overleftarrow{\underline{\rho}} = \overleftarrow{\overline{\rho}} = 0$. This would correspond to not having any spring connecting point $\underline{n}$ to $\underline{x}$ and $\overline{n}$ to $\overline{x}$. The mechanic system would then be indeterminate. When this is the case, we solve the mechanic system in the limit when $\overleftarrow{\underline{\rho}} = \overleftarrow{\overline{\rho}} \to 0^+$. In terms of software implementation, this is achieved by replacing $\overleftarrow{\underline{\rho}}$ and $\overleftarrow{\overline{\rho}}$ by small equal values.

In Section I we explain how to compute the minimum energy configuration of this system quickly. In other words, we show that the minimizer $g^{\text{wall}}$ can be implemented efficiently.

## E  Energy minimizer

The energy minimizer solves the quadratic optimization problem

$$\min_{\{\underline{x},\overline{x}\}} \left[ C\|\underline{x} - \overline{x}\|^2 + (\overleftarrow{\underline{\rho}}/2)\|\underline{x} - \underline{n}\|^2 + (\overleftarrow{\overline{\rho}}/2)\|\overline{x} - \overline{n}\|^2 \right].$$

From the first order optimality conditions we get $2C(\underline{x} - \overline{x}) + \overleftarrow{\underline{\rho}}(\underline{x} - \underline{n}) = 0$ and $2C(\overline{x} - \underline{x}) + \overleftarrow{\overline{\rho}}(\overline{x} - \overline{n}) = 0$. Solving for $\underline{x}$ and $\overline{x}$ we obtain,

$$\underline{x} = \frac{\overleftarrow{\underline{\rho}}\,\overleftarrow{\overline{\rho}}\,\underline{n} + 2C(\overleftarrow{\underline{\rho}}\,\underline{n} + \overleftarrow{\overline{\rho}}\,\overline{n})}{2C(\overleftarrow{\underline{\rho}} + \overleftarrow{\overline{\rho}}) + \overleftarrow{\underline{\rho}}\,\overleftarrow{\overline{\rho}}}, \quad \overline{x} = \frac{\overleftarrow{\underline{\rho}}\,\overleftarrow{\overline{\rho}}\,\overline{n} + 2C(\overleftarrow{\underline{\rho}}\,\underline{n} + \overleftarrow{\overline{\rho}}\,\overline{n})}{2C(\overleftarrow{\underline{\rho}} + \overleftarrow{\overline{\rho}}) + \overleftarrow{\underline{\rho}}\,\overleftarrow{\overline{\rho}}}. \tag{22}$$

If the energy minimizer receives $\overleftarrow{\underline{\rho}} = \overleftarrow{\overline{\rho}} = 0$, we resolve the indeterminacy in computing (22) by letting $\overleftarrow{\underline{\rho}} = \overleftarrow{\overline{\rho}} \to 0^+$.

## F  Maximum velocity minimizer

This minimizer solves the convex problem $\text{minimize}_{\{\underline{x},\overline{x}\}} \left[ (\overleftarrow{\underline{\rho}}/2)\|\underline{x} - \underline{n}\|^2 + (\overleftarrow{\overline{\rho}}/2)\|\overline{x} - \overline{n}\|^2 \right]$ subject to $\|\underline{x} - \overline{x}\| \le C$. If $\|\underline{n} - \overline{n}\| \le C$ then $\underline{x} = \underline{n}$ and $\overline{x} = \overline{n}$. Otherwise, the constraint is active, and, using the KKT conditions, we have $\overleftarrow{\underline{\rho}}(\underline{x} - \underline{n}) = -\lambda(\underline{x} - \overline{x})$ and $\overleftarrow{\overline{\rho}}(\overline{x} - \overline{n}) = -\lambda(\overline{x} - \underline{x})$ where $\lambda \neq 0$ is such that $\|\underline{x} - \overline{x}\| = C$. Solving for $\underline{x}$ and $\overline{x}$ we get,

$$\underline{x} = \frac{\overleftarrow{\underline{\rho}}(\overleftarrow{\overline{\rho}} + \lambda)\underline{n} + \lambda\overleftarrow{\overline{\rho}}\,\overline{n}}{\overleftarrow{\underline{\rho}}\,\overleftarrow{\overline{\rho}} + \lambda(\overleftarrow{\overline{\rho}} + \overleftarrow{\underline{\rho}})}, \quad \overline{x} = \frac{\overleftarrow{\overline{\rho}}(\overleftarrow{\underline{\rho}} + \lambda)\overline{n} + \lambda\overleftarrow{\underline{\rho}}\,\underline{n}}{\overleftarrow{\underline{\rho}}\,\overleftarrow{\overline{\rho}} + \lambda(\overleftarrow{\overline{\rho}} + \overleftarrow{\underline{\rho}})}. \tag{23}$$

To find the solution we just need to determine $\lambda$. Computing the difference between the above expressions we get,

$$\underline{x} - \overline{x} = \frac{\underline{n} - \overline{n}}{1 + (\frac{1}{\overleftarrow{\rho}} + \frac{1}{\overleftarrow{\rho}})\lambda}. \tag{24}$$

Taking the norm of the right hand side and setting it equal to $C$ we get

$$\lambda = \pm \frac{(\|\underline{n} - \overline{n}\|/C) - 1}{\overleftarrow{\rho}^{-1} + \overleftarrow{\rho}^{-1}}. \tag{25}$$

Now examine equation (24). Starting from an $\underline{n} - \overline{n}$ such that $\|\underline{n} - \overline{n}\| > C$, the fastest way to get to $\underline{x} - \overline{x}$ with $\|\underline{x} - \overline{x}\| = C$ is to increase $\lambda > 0$. Hence, in (25), we should choose the positive solution, i.e.

$$\lambda = \frac{(\|\underline{n} - \overline{n}\|/C) - 1}{\overleftarrow{\rho}^{-1} + \overleftarrow{\rho}^{-1}}. \tag{26}$$

If the maximum velocity minimizer receives $\overleftarrow{\underline{\rho}} = \overleftarrow{\rho} = 0$, we resolve any indeterminacy by letting $\overleftarrow{\underline{\rho}} = \overleftarrow{\rho} \to 0^+$. In software, this is achieved by setting $\overleftarrow{\underline{\rho}}$ equal to $\overleftarrow{\rho}$ equal to some small value.

## G  Minimum velocity minimizer

This minimizer can be computed in a very similar way to the maximum velocity minimizer. If $\|\underline{n} - \overline{n}\| \geq C$, then $\underline{x} = \underline{n}$ and $\overline{x} = \overline{n}$. Otherwise, from the KKT conditions, we again obtain equation (23). The difference $\underline{x} - \overline{x}$ is again the expression (24). Now, however, starting from $\underline{n} - \overline{n}$ such that $\|\underline{n} - \overline{n}\| > C$, the fastest way to get to $\underline{x} - \overline{x}$ with $\|\underline{x} - \overline{x}\| = C$, is to decrease $\lambda < 0$. Hence, in (25), we should choose the negative solution, i.e. (26) holds again. If the minimum velocity minimizer receives $\overleftarrow{\underline{\rho}} = \overleftarrow{\rho} = 0$, we resolve any indeterminacy by letting $\overleftarrow{\underline{\rho}} = \overleftarrow{\rho} \to 0^+$. In software, this is achieved by setting $\overleftarrow{\underline{\rho}}$ equal to $\overleftarrow{\rho}$ equal to some small value.

## H  Velocity obstacle minimizers

In this section we explain how to write the minimizers associated to each of the terms in equation (13) using the minimizers $g^{\text{coll}}$, $g^{\text{wall}}$ and $g^{\text{cost}}$ for global planning.

First however, we briefly describe the bipartite graph that connects all these minimizers together. The bipartite graph for this problem has a $g^{\text{VO coll}}$ minimizer-node connecting every pair of equality-nodes. There is one equality-node per variable in $\{x_i\}$. Recall that each of these variables describes the position of an agent at the end of a planning epoch. Each equality-node is also connected to a separate $g^{\text{VO cost}}$ minimizer-node. Finally, for obstacles in 2D, every equality node is also connected to several $g^{\text{VO wall}}$ minimizer-nodes, one per obstacle.

We start by describing the minimizer associated to the terms $\{f^{\text{coll}}_{r_i,r_j}(x_i(0), x_i, x_j(0), x_j)\}$ in equation (13). This is given by

$$g^{\text{VO coll}}(\underline{n}, \underline{n}', \overleftarrow{\underline{\rho}}, \overleftarrow{\underline{\rho}}', x^0, x'^0, r, r') = \arg \min_{\{\underline{x}, \underline{x}'\}} \left[ f^{\text{coll}}_{r,r'}(x^0, \underline{x}, x'^0, \underline{x}') + \frac{\overleftarrow{\underline{\rho}}}{2}\|\underline{x} - \underline{n}\|^2 + \frac{\overleftarrow{\underline{\rho}}'}{2}\|\underline{x}' - \underline{n}'\|^2 \right].$$

The messages $\underline{n}$ and $\underline{n}'$, and corresponding certainty weights $\overleftarrow{\underline{\rho}}$ and $\overleftarrow{\underline{\rho}}'$, come from the equality-nodes associated to the end position of two agents of radius $r$ and $r'$ that, during one time epoch, move from their initial positions $x^0$ and $x'^0$ to $\underline{x}$ and $\underline{x}'$ without colliding.

The outgoing weights $\overrightarrow{\rho}$ and $\overrightarrow{\rho'}$ associated to the variables $\underline{x}$ and $\underline{x}'$ are determined in the following way. If an agent of radius $r$ moving from $x^0$ to $\underline{n}$ does not collide with an agent of radius $r'$ moving from $x'^0$ to $\underline{n}'$, the minimizer will not propose a new trajectory for them, i.e., the minimizer will return $\underline{x} = \underline{n}$ and $\underline{x}' = \underline{n}'$. Hence, in this case, we set all outgoing weights equal to 0, signaling to neighboring equality-nodes that the minimizer wants to have no say when try to reach consensus. Otherwise, we set all outgoing weights equal to $\rho_0$.

For this minimizer, by direct substitution one sees that,

$$g^{\text{VO coll}}(\underline{n}, \underline{n}', \overleftarrow{\underline{\rho}}, \overleftarrow{\underline{\rho}}', x^0, x'^0, r, r') = g^{\text{coll}}(x^0, \underline{n}, x'^0, \underline{n}', +\infty, \overleftarrow{\underline{\rho}}, +\infty, \overleftarrow{\underline{\rho}}', r, r'). \qquad (27)$$

Above we are using a notation where, given a function $f$, $f(+\infty) \equiv \lim_{x \to +\infty} f(x)$. In software, this is implemented by replacing $+\infty$ by a very large value.

In a very similar way, the minimizer associated to the terms $\{f^{\text{wall}}_{x_{Rk}, x_{Lk}, r_i}(x_i(0), x_i)\}$ can be written using the agent-obstacle minimizer for the global planning problem. Concretely,

$$g^{\text{VO wall}}(\underline{n}, x_L, x_R, r, \overleftarrow{\underline{\rho}}) = g^{\text{wall}}(x^0, \underline{n}, r, x_L, x_R, +\infty, \overleftarrow{\underline{\rho}}). \qquad (28)$$

For this minimizer, the rule to set the outgoing weights is the following. If an agent of radius $r$ can move from $x^0$ to $\underline{n}$ without colliding with the line segment $[x_L \, x_R]$ then set all outgoing weights to $0$. Otherwise set them to $\rho_0$.

Finally, we turn to the the minimizer associated to the terms $\{f^{\text{cost}}_{C'_i}(x_i, x_i^{\text{ref}})\}$. This minimizer receives as input a message $\underline{n}$, with corresponding certainty weight $\overleftarrow{\underline{\rho}}$, from the equality-minimizer associated to the position of an agent at the end of a time epoch and outputs a local estimate, $\underline{x}$, of its position at the end of the epoch. It also receives as parameter a reference position $x^{\text{ref}}$ and a cost $c$ of $\underline{x}$ deviating from it. To be concrete, its output is chosen uniformly at random from following arg min set,

$$g^{\text{VO cost}}(\underline{n}, \overleftarrow{\underline{\rho}}, x^{\text{ref}}, c) = \arg\min_{\underline{x}} \left[ f_c^{\text{cost}}(\underline{x}, x^{\text{ref}}) + \frac{\overleftarrow{\underline{\rho}}}{2} \|\underline{x} - \underline{n}\|^2 \right] \qquad (29)$$

$$= \arg\min_{\underline{x}} \left[ c \|\underline{x} - x^{\text{ref}}\|^2 + \frac{\overleftarrow{\underline{\rho}}}{2} \|\underline{x} - \underline{n}\|^2 \right]. \qquad (30)$$

The outgoing weights for this minimizer are always set to $\rho_0$.

Again by direct substitution we see that,

$$g^{\text{VO cost}}(\underline{n}, \overleftarrow{\underline{\rho}}, x^{\text{ref}}, c) = g^{\text{cost}}\left(x^{\text{ref}}, \underline{n}, +\infty, \overleftarrow{\underline{\rho}}, c\right), \qquad (31)$$

where in $g^{\text{cost}}$ we are using the energy minimizer for the global planning problem.

# I  Mechanical analog

In this section we explain how to compute the minimum energy configuration of the springs-slab system described in Section D. Basically, it reduces to computing the minimum of a one-dimensional real function over a closed interval.

Given $\underline{n}$ and $\overline{n}$, two main scenarios need to be considered.

1. If $\underline{n}, \overline{n} \notin \mathcal{R}$ and $[\underline{n} \, \overline{n}] \cap \mathcal{R} = \emptyset$, i.e. the segment from $\underline{n}$ to $\overline{n}$ does not intersect $\mathcal{R}$, then $\underline{x} = \underline{n}$ and $\overline{x} = \overline{n}$.

2. Otherwise, because there might be multiple local minima, i.e. multiple stable static configurations, we need to compare the energy of the following two configurations and return the one with minimum energy.

   (a) The slab is tangent to $\mathcal{R}$, for example as in the blue configuration of Figure 6-right.
   (b) One of the springs is fully compressed and exactly one end of the slab is touching the boundary of $\mathcal{R}$, for example as in the red configuration of Figure 6-right.

Let us compute the energy for Scenario 2a. The slab can be tangent to $\mathcal{R}$ in many different ways. However, the arrangement must always satisfy two properties. First, the point of contact, $p$, between the slab and $\mathcal{R}$ touches either the boundary of the semi-circle centered at $x_L$ or the boundary of the semi-circle centered at $x_R$. Second, because extending/compressing the slab costs zero energy, the slab must be orthogonal to the line segment $[\underline{n} \, \underline{x}]$ and to the line segment $[\overline{n} \, \overline{x}]$.

The first observation allows us to express $p$ using the map $P(\theta) : [0, 2\pi] \mapsto \mathbb{R}^2$ between the direction of the slab and the point of contact at boundary of the semi-circles,

$$P(\theta) = \begin{cases} x_R + r\hat{n}(\theta) & \langle x_R - x_L, \hat{n}(\theta) \rangle \geq \langle x_L - x_R, x_R \rangle \\ x_L + r\hat{n}(\theta) & \text{otherwise} \end{cases} \tag{32}$$

where $\hat{n}(\theta) = \{\cos(\theta), \sin(\theta)\}$. Specifically, there is a $\theta^0 \in [0, 2\pi]$ such that $p = P(\theta^0)$. The second observation tells us that $\underline{x} = \underline{n} + \underline{\gamma}\hat{n}(\theta^0)$ and $\overline{x} = \overline{n} + \overline{\gamma}\hat{n}(\theta^0)$ where $\underline{\gamma}$ and $\overline{\gamma}$ can be determined using the orthogonality conditions,

$$\langle \underline{x} - P(\theta^0), \hat{n}(\theta^0) \rangle = 0 \Rightarrow \underline{\gamma} = \langle P(\theta^0) - \underline{n}, \hat{n}(\theta^0) \rangle \tag{33}$$

$$\langle \overline{x} - P(\theta^0), \hat{n}(\theta^0) \rangle = 0 \Rightarrow \overline{\gamma} = \langle P(\theta^0) - \overline{n}, \hat{n}(\theta^0) \rangle. \tag{34}$$

Therefore, the minimum energy configuration over all tangent configurations, which is fully determined by $\theta^0$, must satisfy

$$\theta^0 \in \arg\min_{\theta \in [0, 2\pi]} E_{\text{tangent}}(\theta) \qquad \text{where,} \tag{35}$$

$$E_{\text{tangent}}(\theta) = \frac{\overleftarrow{\underline{\rho}}}{2}(\langle P(\theta) - \underline{n}, \hat{n}(\theta) \rangle)^2 + \frac{\overleftarrow{\overline{\rho}}}{2}(\langle P(\theta) - \overline{n}, \hat{n}(\theta) \rangle)^2. \tag{36}$$

Problem (35) involves minimizing the one-dimensional function (36). Figure 7-left shows the typical behavior of $E_{\text{tangent}}(\theta)$. It is non-differentiable in at most 2 points. When the agent-obstacle

Figure 7: Typical behavior of $E_{\text{tangent}}(\theta)$ for the agent-obstacle minimizer (left) and for the agent-agent minimizer (righ).

minimizer is used to solve the agent-agent minimizer, the function becomes smooth and has second derivative throughout all its domain, see Figure 7-right. In the numerical results of Section 4, our implementation of the agent-agent minimizer uses Newton's method to solve (35). To find the global minimum, we apply Newton's method starting from four equally-space points in $[0, 2\pi]$. To produce the video accompanying this appendix, our implementation of the agent-obstacle minimizer solves (35) by scanning points in $[0, 2\pi]$ with a step size of $2\pi/1000$. In this case, it is obvious there is room for improvement in speed and accuracy by choosing smarter ways in which to solve (35).

To compute the energy for Scenario 2b, we need to determine which of the springs is fully contracted, or which side of the slab is touching $\mathcal{R}$. If $\underline{n} \in \mathcal{R}$ and $\overline{n} \notin \mathcal{R}$ then $\overline{x} = \overline{n}$ and $\underline{x}$ is the point in the boundary of $\mathcal{R}$ closest to $\underline{n}$ such that $[\underline{x}\,\overline{x}]$ does not intersect $\mathcal{R}$. Since the boundary of $\mathcal{R}$ is formed by parts of the boundary of two circles and of two lines, this closest point can be computed in closed form. If $\overline{n} \in \mathcal{R}$ and $\underline{n} \notin \mathcal{R}$ the situation is the opposite. If $\underline{n}, \overline{n} \in \mathcal{R}$, then we know we cannot be in Scenario 2b. Finally, if $\underline{n}, \overline{n} \notin \mathcal{R}$, we compute the energy assuming that $\underline{x} = \underline{n}$ and then assuming that $\overline{x} = \overline{n}$ and take the configuration with smallest energy between them.

## J  Comment on our algorithm

Note that our algorithm does *not* possess anytime guarantees and, if stopped earlier, the trajectories might have collisions. However, if stopped early, a suboptimal set of non-colliding trajectories can be found at very low computational cost by using our algorithm to solve the feasibility problem in (2) starting from the state of the algorithm at stop time. In addition, although dynamic/static obstacles

can be seamlessly integrated into our framework, a solution must be recomputed (as is the case with A* or RRT*) if their trajectories/positions change unexpectedly. This being said, in our algorithm, if a new piece of information is received, the previous solution can be used as the initial guess, potentially decreasing convergence time. Note that in some scenarios, a low-cost local-planning approach, such as the one presented in Section 5, can be beneficial.

## K    Comment on the scaling of convergence time with $p$

Based on [13], we think that in non-adversarial configurations, in contrast to CONF1 where dead-locks are likely, the scaling of convergence time with $p$ is not exponential. Our reasoning is based on a connection between trajectory planning and disk packing. For example, minimum-energy trajectory planning using piece-wise linear trajectories is related to, although not the same as, packing disks in multiple 2D layers, where the two matched disks between consecutive layers generate a larger cost when far away from each other. The numerical results in [13] report that, for disk packing, the runtime of ADMM and TWA is no more than polynomial in the number of disks and we believe the runtime for trajectory planning for non-adversarial configurations has a similar complexity. We interpret the seemingly exponential curve of convergence time versus $p$ for $n = 8$ in Figure2-left as an atypical, adversarial scenario. By comparison, in Figure 3-left, which assumes randomly chosen initial and final points and also minimization of energy, the dashed-blue curve of runtime versus $p$ for $n = 8$ does not appear to exhibit exponential growth.

## Footnotes

[3]ADMM is a decomposition procedure for solving optimization problems. It coordinates the solutions to small local sub-problems to solve a large global problem. Hence, it is useful to derive parallel algorithms. It was introduced in [16] and [17] but is closely related to previous work as the dual decomposition method introduced by [18] and the method of multipliers introduced by [19, 20] and [21]. For a good review on ADMM see [22], where you can also find a self-contained proof of its convergence for convex problems.