[Reviews · NeurIPS 2013]

Submitted by Assigned_Reviewer_4

The paper considers global and local path planning for multiple agents in 2-D with a centralized message-passing algorithm derived from the "three-weight" version of ADMM, an established algorithm. The contributions are clearly stated in the introduction: The authors decompose global planning optimization into several sub-problems they dub "minimizers," which describe various planning objectives that comprise the larger overall problem to be solved. Minimizers are derived for avoiding inter-agent collisions, avoiding collisions with static obstacles, and for maximizing/minimizing kinetic energy or velocity. They also apply their approach to local planning by reformulating joint optimization.

The overall concept for using a distributed optimization algorithm for multi-agent path planning is nice; however the
results are a little lacking. They do not provide comparisons to other well-known high dimensional path planning techniques,
such as A* variants with inflated heuristics or sampling-based methods. For many of these problems, there is also the aspect of replanning given new state or obstacle information. Would this algorithm be amenable to such dynamic or anytime implementations? In that regard, it's hard to see how this formulation would guarantee that certain constraints would always be enforced for early stopping, for example a hard collision or communication constraints. These and other issues regarding the planning problem should be discussed. It's also not clear if this manuscript would appeal to the broader NIPS audience. It would definitely fit into a dedicated planning/robotics meeting, but it's hard to see if this would appeal from an algorithmic viewpoint to other NIPS areas.

There are also a few minor items that could be improved in the presentation. First of all, please use a letter other than "n" for the correction messages in your formulations. "n" is already used to describe the number of interior break-points, a major parameter for the overall algorithm. Figure 2-middle and Figure 3-left include multiple scales on either the x- or y-axis. In the case of Figure 3-left, I initially thought that the convergence time between optimizing total energy and finding a feasible solution were comparable, when one is several orders of magnitude larger. These figures seem to be an attempt to fit the paper within the page limit but could cause confusion and dilute the impact of the results. Also, with respect to the convergence times reported for local trajectory planning using the proposed approach versus MIQP [12], the authors compare implementations in different programming languages and say the results are not "exactly comparable." Yet, they directly compare the convergence times in Figure 3-right. They also state, "our method does not require approximating the search domain by hyperplanes, nor does it require an additional branch-and-bound algorithm," which causes one to further question the direct comparison of convergence times. These results could be clarified.
Summary: Interesting application of distributed message-passing approach to optimization for multi-agent path planning.

Submitted by Assigned_Reviewer_5

This paper proposes a message-passing algorithm for multi-agent trajectory planning. It formulates the planning problem as a numeric optimization in which the speed and direction of each agent at each "break point" in time are the variables, and minimizing the energy required and avoiding collisions are the objective. This optimization is solved via the three-weights variant of the alternating direction method of multipliers (ADMM), where ADMM is used to break up the objective with a consensus-optimization pattern. This paper derives solutions to the specific subproblems induced by ADMM. The algorithm is evaluated on standard mutlti-agent trajectory planning tasks, and compares favorably with the original version of ADMM.

This paper thoroughly examines the problem of multi-agent trajectory planning and introduces a novel solution. It derives minimizers for several useful objective terms: energy minimization, velocity maximization, velocity minimization, agent-agent collision avoidance, and agent-wall collision avoidance. The empirical results are very convincing. The three-weights variant is significantly better than the original version of ADMM. The proposed algorithm scales well to high numbers of agents. It also often finds good solutions to the problems. Is it correct to assume that the global optimum is a little less than 300 in the middle of Figure 2? If so, then the solutions are usually within 33%. Further, the solutions shown in the supplementary material are indeed graceful.

Minor points:
--- At the end of Section 2, the contrast between ADMM and the three-weights variant could be improved. The intuitions given in point 3 of the algorithm description in the supplementary material are much better, and the authors should consider adding some of that material to the main paper.
--- Boyd et al., citation 21, should be cited as a book in a series, not a journal article.
Summary: This paper introduces a three-weight alternating direction method of multipliers (ADMM) approach to multi-agent trajectory planning. The proposed solution is compelling and performs well.
Author Feedback

Author rebuttal: We thank the reviewers for their valuable feedback.

-- To all reviewers
We will fix all the identified typos and try to incorporate all suggestions regarding presentation and organization.

Regarding the originality of our work, we stress that the primary contribution of this paper is not an incremental improvement upon previous trajectory-planning algorithms, but instead the introduction of a novel, unified framework that subsumes and expands upon the desirable properties of many previous approaches. Namely, it (i) jointly optimizes full continuous trajectories for all agents, (ii) is easily parallelizable, (iii) can flexibly account for static/dynamic obstacles and restrictions on energy and velocity, (iv) and has a message-passing interpretation that lends to decentralized implementations.

In addition, given the rising interest in the NIPS community in distributed optimization, and in particular ADMM, we believe our work will have an impact beyond the areas of Robotics and Multi-Agent systems. Although there exist applications of ADMM to non-convex problems, these only consider objective functions that can be decomposed into a few convex/concave sub-problems (e.g. Zhang et. al 09 or [21] Chapter 9; typically applied to matrix factorization/completion and non-linear data-fitting with regularization). As far as we know, our work is the first to show a successful application of ADMM to a real-world application where the objective is decomposed into a very large number of non-convex sub-problems. A notable exception is recent work in [13], although the applications were primarily of academic interest. This success could have a major impact on the part of NIPS community interested in large-scale optimization.

-- To Reviewer 7
We thank the reviewer for suggesting a simple baseline algorithm. We agree it can clarify if our performance comes from our specific optimization formulation, from our ADMM-based algorithm, or both. We will certainly test it.

Regarding how the number of agents affects the runtime: based on [13], we believe that in non-adversarial configurations, in contrast to CONF1 where dead-locks are likely, the complexity is not exponential. Our belief is based on a connection between trajectory planning and disk packing. For example, minimum-energy trajectory-planning using piece-wise linear trajectories is related to, although not the same as, packing disks in multiple 2D layers, where the two matched disks between consecutive layers generate a larger cost when far away from each other. The numerical results in [13] report that, for disk packing, the runtime of ADMM and TW is no more than polynomial in the number of disks and we believe the typical runtime for trajectory planning has a similar complexity. We interpret the seemingly exponential curve of convergence time vs p for n = 8 in Fig.2 as an atypical, adversarial scenario. By comparison, in Fig.3, which assumes randomly chosen initial and final points and also minimization of energy, the dashed-blue curve of runtime vs p for n = 8 does not appear to exhibit exponential growth.

-- To Reviewer 5
We cannot be completely certain that the optimal solution has a value of around 300. It is conceivable that the optimal solution is at the bottom of a deep and narrow well of the objective function that is hard to reach even by randomly initializing the algorithm. This being said, looking at the behavior of the trajectories found, we are inclined to agree that, for the experiment associated to Fig.2-middle, the optimum should be around that value and hence that the dispersion is about 33%.

-- To Reviewer 4
We agree we did not discuss some aspects of trajectory planning, including a more detailed comparison to other methods. Unfortunately, space limitations required us to make difficult choices of what to include. This discussion is left as future work.

Our algorithm does not possess anytime guarantees and, if stopped earlier, the trajectories might have collisions. However, if stopped early, a suboptimal set of non-colliding trajectories can be found at very low computational cost by using our algorithm to solve the feasibility problem in Eq. (2) starting from the state of the algorithm at stop time.

Dynamic/static obstacles can be seamlessly integrated into our framework, but a solution must be recomputed (as is the case with A* or RRT*) if their trajectories/positions change unexpectedly. This being said, in our algorithm, if a new piece of information is received, the previous solution can be used as the initial guess, potentially decreasing convergence time. Note that in some scenarios, a low-cost local-planning approach, such as the one presented in Section 5, can be beneficial.

A*-search based methods and sampling-based methods require exploring a continuous domain using discrete graph structures. For large problems, fixed-grid search methods are impractical. Alternatively, exploration can be done using random/sampling algorithms with proved asymptotic convergence to optimality, Karaman et al., 2010. However, as the dimensionality of the configuration space increases, the convergence rate degrades and the local planners required by the exploration loop become harder to implement. In addition, as pointed out in [9], sampling algorithms cannot easily produce solutions where multiple agents move in tight spaces, like in CONF1 with obstacles. Some of the disadvantages of using discrete random search structures are even visible in extremely simple scenarios. For example, for a single holonomic agent that needs to move as quickly as possible between two points in free-space, Karaman et al., 2010, require around 10000 samples on their RRT* method to find something close to the shortest-path solution. For our algorithm this is a trivial scenario: it outputs the optimal straight-line solution in 200 iterations and 37 msecs. in our Java implementation.

These points will be clarified in the text.